# *Dirofilaria* sp. and Blood Meal Analysis in Mosquitoes Collected in Vojvodina and Mačva, and the First Report of *Setaria tundra* (Issaitshikoff & Rajewskaya, 1928) in Serbia

**DOI:** 10.3390/ani14091255

**Published:** 2024-04-23

**Authors:** Sara Šiljegović, Théo Mouillaud, Davy Jiolle, Dušan Petrić, Aleksandra Ignjatović-Ćupina, Ana Vasić, Christophe Paupy, Mihaela Kavran

**Affiliations:** 1Faculty of Agriculture, Centre of Excellence—One Health, University of Novi Sad, Trg Dositeja Obradovića 8, 21000 Novi Sad, Serbia; sarasiljegovic@yahoo.com (S.Š.); dusanp@polj.uns.ac.rs (D.P.); cupinas@polj.uns.ac.rs (A.I.-Ć.); 2Maladies Infectieuses et Vecteurs: Écologie, Génétique, Évolution et Contrôle (MIVEGEC), Montpellier University, Institut de Recherche pour le Développement (IRD), Centre National de la Recherche Scientifique (CNRS), 34090 Montpellier, France; theo.mouillaud@ird.fr (T.M.); davy.jiolle@ird.fr (D.J.); christophe.paupy@ird.fr (C.P.); 3Scientific Institute of Veterinary Medicine of Serbia, Janisa Janulisa 14, 11000 Belgrade, Serbia; ana.vasic@nivs.rs

**Keywords:** *Dirofilaria* sp., *Setaria tundra*, mosquito surveillance, Cox1 gene

## Abstract

**Simple Summary:**

Parasitic filarial nematodes of the genus *Dirofilaria* pose a significant threat to veterinary health, affecting dogs, cats, and occasionally humans. In Serbia, *Dirofilaria* infections are endemic, with prevalence rates documented in both animals and humans. However, our knowledge about vectors remains limited. Recently, mosquitoes have been identified with positive indications for the presence of *Dirofilaria*. The parasite *Setaria tundra* is a significant parasite of deer in Europe, but it had not been reported in Serbia until now. This research aims to map out *Dirofilaria* hotspots in Vojvodina Province, identify positive mosquito species carrying the nematodes, and analyze blood-fed mosquitoes to determine potential sources of infection. Through collecting and analyzing 2902 female mosquitoes from 73 locations during 2021 and 2022, the study detected *D. immitis* in three locations (Zrenjanin, Glogonj, and Svetozar Miletić) and *Setaria tundra* in two locations (Iđoš and Mali Iđoš). *Dirofilaria immitis* was detected in *Culex pipiens* mosquitoes, and *Setaria tundra* was detected in *Aedes vexans* and *Aedes caspius*, expanding our understanding of nematode distribution in Serbia. Blood meal analysis sheds light on the feeding preferences of infected mosquitoes.

**Abstract:**

*Dirofilaria immitis* and *D. repens* are the two most widespread and important species of mosquito-borne nematodes, posing a significant threat to veterinary health and particularly affecting canines and felines. While *D. immitis* causes cardiopulmonary dirofilariasis, *D. repens* causes subcutaneous infections in dogs and other carnivores. Despite the extensive knowledge on these parasites, little is known about their natural vectors in Serbia. The parasite *Setaria tundra*, known to infect deer, has not yet been detected in Serbia but has been documented in neighboring countries. Thus, the aim of this study was to (i) further map out *Dirofilaria* sp. hotspots in the Vojvodina Province and detect *S. tundra* for the first time, (ii) detect positive mosquito species that can provide insights into how the nematodes spread in Serbia, and (iii) analyze the blood-fed female mosquitoes of species found to be infected, in order to identify the potential source of parasite infection. A total of 2902 female mosquitoes were collected across 73 locations during 2021 and 2022. Molecular biology methods, based on conventional PCR, were used to analyze non-blood-fed (2521 specimens) and blood-fed (381 specimens) female mosquitos, in order to detect filarial nematode presence and identify blood-meal sources, respectively. When the parasite genome was detected, the amplicon (cox1 gene, 650 bp fragment) was sent for Sanger sequencing, further confirming the presence of nematodes and species assignation. *D. immitis* was detected in three *Culex pipiens* mosquitoes collected in Zrenjanin (August 2021) and Glogonj and Svetozar Miletić (both in July 2021). Additionally, *Setaria tundra* was detected in *Aedes vexans* collected in Iđoš (mid-August 2021) and *Aedes caspius*, which was collected in Mali Iđoš (end of July 2021). This work identifies two new locations where *D. immitis* occurs in Vojvodina, and is the first report of *S. tundra* in Serbian territory. Blood-meal analysis provided insights into the preferences of mosquitoes that were positive for *Dirofilaria* sp. and *S. tundra*.

## 1. Introduction

The family *Onchocercidae* (Nematoda: Filarioidea) comprises eight subfamilies, including *Waltonellinae*, *Oswaldofilariinae*, *Icosellinae*, *Splendidofilariinae*, *Lemdaninae*, *Onchocercinae*, and the two main subfamilies of interest in this paper, *Dirofilariinae* and *Setariinae* [1]. These subfamilies contain parasitic filarial nematodes that infect all vertebrates, excluding fish [2].

The parasitic filarial nematode of the genus *Dirofilaria* represents a severe threat to veterinary and public health, particularly affecting dogs and cats, and, on rare occasions, humans, as well [3,4,5,6]. Besides canines and felines, these cosmopolitan parasitic worms [7] might also infect other carnivores, such as wolves (*Canis lupus*), red foxes (*Vulpes vulpes*), and golden jackals (*Canis aureus*) [8,9,10,11].

*Dirofilaria immitis* (Leidy 1856), an important mosquito-borne nematode, known as the dog heartworm, causes cardiopulmonary dirofilariasis, invading the heart and large blood vessels [12]. The damage caused by this parasite to the arteries and right cardiac chambers of infected hosts might have a fatal outcome, especially if not treated or if treatment is delayed. Another dirofilarial worm is *Dirofilaria repens* Railliet et Henry, 1911, which causes subcutaneous infections in dogs and other carnivores [13]. Both *Dirofilaria* species can accidentally be transmitted to humans [14,15,16,17]. Although humans are dead-end hosts to these filarial nematodes (as they cannot proliferate in the human body), they can still cause health issues, depending on the invaded body part. The infection may manifest superficially, with the adult nematodes appearing subcutaneously and subconjunctivally [18]. However, the major concern in human populations is represented by the benign pulmonary nodules caused by *D. immitis* in the human lungs, frequently mistaken for malignant lung tumors [15,19,20,21].

The nematodes of the genus *Setaria* Viborg, 1795 are parasites of different ungulates, including artiodactyls, equines, and even African hyraxes [2]. Their main vectors of transmission are mosquitoes belonging to the Culicidae family and flies from the Simuliidae and Muscidae families [22]. Typically, *Setaria* species do not cause clinical disease and thus often remain undetected, but they may cause mild chronic peritonitis in mammalian hosts [2].

*Setaria tundra* (Issaitshikoff & Rajewskaya, 1928) has recently attracted interest in new studies [23,24,25] due to its expanding geographical range towards southern Europe and its negative impact on wild and semi-domesticated reindeer. Studies that performed the xenomonitoring of mosquitoes for the presence of *Dirofilaria* spp. typically detected *S. tundra* alongside it [22,26,27].

Nowadays, cases of dirofilarial infections have been detected worldwide [3]. The process of parasite transmission to hosts is very complex. Successful transmission requires the presence of competent mosquito vectors. Once a mosquito female intakes blood infected with microfilariae, in the following two or more weeks, nematodes molt to the infective third larval stage. The infective stage moves from the tubules via the hemocoel to the lumen of the labial sheath in the mosquito’s mouthparts [28]. The duration of this period, measured in the body of several mosquito species (*Aedes vexans* (Meigen 1830), *Aedes triseratus* Say 1823, *Aedes trivittatus* (Coquillett 1902), and *Anopheles quadrimaculatus* Say 1824), lasts 14 days and is directly temperature-dependent [29,30,31,32]. The subsequent blood meal intake of an infected female mosquito will result in parasite transmission to the bitten host [12,15].

Regardless, around 70 mosquito species classified into the *Anopheles*, *Aedes*, *Culex*, *Culiseta*, and *Coquillettidia* genera have been considered as potential vectors of animal and human dirofilariasis, whereas only a few species have been proven to be competent vectors [12,33]. 

Serbia has been considered an endemic country of *Dirofilaria* sp. in animals and humans for many years [12]. Several studies have been conducted targeting *Dirofilaria* in reservoirs (animals) and humans [34,35,36,37,38,39,40]. Between 2006 and 2007, the reported prevalence for *D. immitis* in dogs was 7.2% in the Vojvodina region and 3.2% in the Branicevo region [34,35]. In the region of Belgrade, a few years later, the prevalence of *D. immitis* in dogs was 22.01% and 3.97% of dogs were co-infected with *D*. *repens* [12].

Despite all the knowledge on the presence of *Dirofilaria* sp. in Serbia, little is known about their vectors. So far, only one publication has focused on the vectors of *Dirofilaria* [41]. Kurucz et al. [41] showed that 8.3% of tested mosquito pools were positive for *Dirofilaria*. Positive mosquitoes belonged to five mosquito species: *Ae. vexans*, *Aedes caspius* (Pallas 1771), *Aedes sticticus* (Meigen 1835), *Culex pipiens* Linnaeus 1758, and *Coquillettidia richiardii* (Ficalbi 1889). Mosquitoes were found positive for both *D. immitis* and *D. repens* at several localities throughout the entire mosquito breeding season. 

The aim of the present study is to contribute to the mapping of *Dirofilaria* hotspots and report, for the first time, the presence of *S. tundra* in Vojvodina Province and the Mačva region, Serbia. Detecting positive mosquitoes can provide insights into the parasite’s distribution in Serbia, helping us understand its spread in the region. Analyzing the blood meals of vectors could help us to create a list of animal species that may be at risk due to potential *Dirofilaria* infections. 

## 2. Materials and Methods

### 2.1. Mosquito Sampling and Vector Identification

Mosquito sampling was conducted in Vojvodina Province, Serbia (65 locations), covering an area of 21,506 km^2^. In addition, eight locations belonging to the Mačva region (612 km^2^) were included. The locations were selected based on two criteria: first, to evenly cover the entire territory of Vojvodina Province and Mačva, andsecond, to prioritize areas with the presence of domestic animals and humans, where possible. Sampling was carried out at 73 locations in total (Figure 1) during the summer season of mosquito activity in 2021, from May to October. Due to the low number of *Aedes albopictus* Skuse 1894 collected in 2021 and the high significance of the filarial transmission of this invasive species, we included the samples from 2022 to increase the likelihood of parasite detection. The geo-coordinates of locations are shown in Appendix A. This study only included adult female mosquitoes. Females were collected using CO_2_-baited (dry ice) adult traps (NS2 trap type). Traps were set up in the afternoon hours and operated overnight. Mosquito samples were then kept in dry ice until being transferred to the laboratory within the Centre of Excellence—One Health at the Faculty of Agriculture, University of Novi Sad, Serbia. When the samples arrived at the laboratory, mosquitoes were morphologically identified to species level, using the identification key by Becker et al. [42]. 

All collected females per location were categorized based on the presence of blood meal in their abdomen as non-blood-fed or blood-fed. Females were separated into pools of up to 100 individuals per species per tube. From each mosquito trap, only one pool per species was taken. Samples were conserved dry in 2 mL tubes (Eppendorf, Hamburg, Germany) and stored in a freezer at −20 °C until being analyzed.

Due to the regularly high number (>200 per trap) of non-blood-fed mosquitoes in traps in the majority of locations, a selection of mosquito species (for further analysis) from this category was based on vector competence to transmit *Dirofilaria* sp. Selected mosquito species were *Ae. vexans*, *Ae. caspius*, *Ae. albopictus*, and *Anopheles maculipennis* Complex Meigen 1818.

The number of blood-fed females in traps was usually very low (<5 per trap); therefore, we analyzed all captured blood-fed mosquito species for the presence of *Dirofilaria* sp. Because of this low number of blood-fed specimens, we also included mosquitoes collected in 2022. 

After screening non-blood-fed and blood-fed mosquitoes for the presence of parasites, we analyzed the blood meal source in blood-fed females to identify the putative host species. The following selection for host detection included (a) females from the positive locations belonging to the same species as the positive ones and (b) females from locations in the close vicinity to the positive locations. Additionally, non-blood-fed females that belonged to the same species and same locations (referring to a and b from above) were also added to try to detect the host (it was assumed that some females might have already digested a blood meal and it was not visible in the abdomen).

### 2.2. DNA Extraction

Extractions and the molecular analyses of all samples were conducted at the Institute of Research and Development, within the Mivegec research unit, Montpellier, France. 

Extraction of DNA was carried out by using the DNeasy Blood and Tissue Kit (Qiagen, Hilden, Germany), according to the manufacturers’ instructions. 

For parasite detection, non-blood-fed mosquitoes were pooled in tubes of up to 20 individuals for DNA extraction. Therefore, pools with a number of mosquitoes that was higher than 20 had to be divided. Meanwhile, for blood-fed females, we distributed one mosquito per tube for further analysis (if positive) for blood-meal source detection.

Positive controls of *D. repens* and *D. immitis* were extracted from infected dogs’ blood and were provided by Dr. Ettore Napoli (University of Messina, Department of Veterinary Sciences). DNA extraction of positive controls was also performed using the Dneasy Blood and Tissue Kit.

### 2.3. Identification of Dirofilaria sp.

Screening of mosquito pools for the presence of *Dirofilaria* sp. was conducted using a conventional PCR approach based on the amplification of the cytochrome oxidase I (COI or cox1) gene in parasites. The cox1 gene was targeted using the primer pair COIintF (5′-TGATTGGTGGTTTTGGTAA-3′) and COIintR (5′-ATAAGTACGAGTATCAATATC-3′) under the modified PCR conditions described by Casiraghi et al. [43,44], Gabrielli et al. [45], and Tasić-Otašević et al. [46].

Polymerase chain reaction (PCR) was performed in 25 µL volumes of mixture under the following final conditions: 16.05 µL of water, Tp 10× 2.5 µL (Eurogentec, Seraing, Belgium) including 50 mM MgCl_2_ 0.75 µL (Eurogentec), 10 mM dNTP 0.5 µL (Eurogentec), primer COI-int-F (10 pmol/µL = 10 µM) 1.5 µL, primer COI-int-R (10 pmol/µL = 10 µM) 1.5 µL, and TAQ Platinum (5 U/µL) 0.2 µL (Invitrogen, Waltham, MA, USA). Two µL of sample DNA was added to 23 µL of Master mix.

The thermal profile used was 94 °C for 10 min and then 5 cycles at 94 °C for 30 s, 52 °C for 45 s, and 72 °C for 1 min, then, afterward, 30 cycles at 94 °C for 30 s, 58 °C for 45 s, and 72 °C for 1 min. The final cycle was at 72 °C for 7 min. These conditions provided PCR products of 650 bp. 

PCR products were separated by TAE 0.5× and 1.3% agarose gel electrophoresis (Eurogentec) stained with gelred (Biotium, San Francisco, CA, USA) and sized with 4.5 µL ladder (Generuler 100 bp, Thermo Scientific, Waltham, MA, USA). The quantity used for the preparation of gel was as follows: 50 mL of TAE 0.5×, 0.65 g of agarose and 10 µL of stain gelred. The product was then migrated for 35 min at 100 V.

Samples which produced bands were further processed via Sanger sequencing (Eurofins Genomics, Konstanz, Germany). Assembled sequences were subjected to NCBI nucleotide BLAST tool (blastn). The search set was configured with standard database parameters, while, for the program selection, the highly similar sequence (megablast) was selected. Results of the BLAST analysis showing only the highest percent identity (98–100%) were considered in this study.

The consensus sequences were produced and cleaned in BioEdit (version 7.7.1). Sequence alignment was performed using the ClustalW method. The same protocol was performed for the three samples that were positive for *D. immitis*, as well as the two samples that were positive for *S. tundra*, with the aim of validating previously detected parasites.

### 2.4. Identification of Blood-Meal Host

Molecular identification of blood-meal source species was performed following the protocol by Boessenkool et al. [47]. The primers used were 16Smam1 (CGGTTGGGGTGACCTCGGA) and 16Smam2 (GCTGTTATCCCTAGGGTAACT). PCR was performed in a final volume of 50 µL under the following conditions: water 36 µL, Tp 10× 5 µL (Eurogentec), MgCl_2_ 50 mM 2 µL (Eurogentec), dNTP 10 mM 0.2 µL (Eurogentec), primer 16Smam1 (10 pmol/µL = 10 µM) 0.8 µL, primer 16Smam2 (10 pmol/µL = 10 µM) 0.8 µL, and TAQ Platinum (5 U/µL) 0.2 µL (Invitrogen). We added 45 µL of Master mix + 5 µL of DNA.

Thermal profile consisted of 55 cycles with the temperatures as follows: 94 °C for 2 min, 94 °C for 30 s, 60 °C for 30 s, 72 °C for 30 s, and 72 °C for 10 min. These conditions provided PCR products of 150 bp.

PCR products were separated using TAE 0.5× and 2% agarose gel electrophoresis stained with gelred and sized with 4.5 µL ladder. The quantities used for the preparation of gel were 50 mL of TAE 0.5×, 1 g of agarose and 10 µL of stain gelred. The product was then migrated for 35 min at 100 V. Amplicons were sent for sequencing to Eurofins.

The assembled sequences were compared with those in GenBank using a nucleotide BLAST tool (blastn). Standard database parameters were used for the search set, and the highly similar sequences (megablast) program was selected.

Regarding the results of the BLAST analysis, only those with the highest percent identity (98–100%) were included in this study.

### 2.5. Phylogenetic Analysis of Setaria tundra

Phylogenetic analysis of *S. tundra* nucleotide sequences (~650 bp fragment) was performed using BLAST NCBI and MEGA v. 11.0 software [48] to align sequences and determine phylogenetic relationships. Maximum Likelihood with the Jones–Taylor–Thornton substitution model was used as the tree construction method. Additionally, BLAST searches were performed in GenBank, https://www.ncbi.nlm.nih.gov, (accessed on the 15 March 2024) and *S. tundra* matches showing a high genetic affinity were downloaded and incorporated into the alignment. Bootstrap analysis of 1000 randomly generated sample trees was performed to assess the stability of the inferred phylogenies. The selected outgroups were *D. immitis* and *Setaria cervi* (Rudolphi, 1819).

## 3. Results

### 3.1. Presence of Dirofilaria immitis and Setaria tundra in Mosquitoes 

The total number of analyzed female mosquitoes was 2902, of which 2521 were non-blood-fed mosquitoes and 381 were blood-fed mosquitoes; eight species were analyzed (Table 1). Out of the 2902 screened mosquitoes, the genome of filaria was found in only five mosquito pools (in total, six mosquitoes; one pool consisted of two mosquitoes). All positive mosquitoes were collected in Vojvodina Province. Mosquitoes from the Mačva region were not positive for the target parasites.

The sequencing and the BLAST analysis confirmed the presence of *D. immitis* in three samples, all of which were detected in *Cx. pipiens* mosquitoes (Figure 2). Positive *Cx. pipiens* were collected in three different locations: Glogonj, Svetozar Miletić, and Zrenjanin. Positive mosquitoes in Glogonj and Svetozar Miletić were collected in July 2021, while in Zrenjanin, *Cx. pipens* was positive at the end of August 2021. *D. immits* was present only in blood-fed *Cx. pipiens*. 

The results also showed that two out of five positive samples were positive for *Setaria tundra*, a species of nematode that has not been detected before in Serbian territory. In this study, *S. tundra* was detected in two mosquito species, *Ae. caspius* and *Ae. vexans* (Figure 2). *Aedes caspius* was collected in the location Mali Iđoš, at the end of July 2021, while *Ae. vexans* was collected at the location Iđoš, during mid-August 2021. This parasite was detected in non-blood-fed mosquitoes.

All five locations with positive mosquitoes are shown in Figure 3.

### 3.2. Blood-Meal Host Detection

Out of five positive locations for parasites, blood-fed females were collected only in four. Besides these four, four additional neighboring locations were included in the analyses. In total, blood-fed females from eight locations were analyzed. 

Out of 30 selected females, 22 were blood-fed females, and eight were non-blood-fed females. We analyzed 19 *Cx. pipiens* (blood-fed), seven *Ae. vexans* (three blood-fed and four non-blood-fed), and four *Ae. caspius* (non-blood-fed).

In total, 16 mosquitoes resulted in successful host detection. One mosquito was non-blood-fed, and the rest of them were blood-fed. The identified hosts are presented in Figure 3. The host was not identified in any of the *Ae. caspius* females analyzed.

### 3.3. Phylogenetic Analysis of Setaria tundra

The approximate 650 bp fragment of the cox1 gene was analyzed in two isolates. *Setaria tundra* isolated from *Ae. caspius* has shown similarity with *S. tundra* originally isolated from *Cq. richiardii* in Austria (MF695090), while *S. tundra* isolated from Ae. vexans has shown a similarity with *S. tundra* isolated from *Ae. vexans* in Hungary (KM452922) (Figure 4).

All new nucleotide sequences in this study have been deposited in GenBank NCBI with the accession numbers PP475177 (*S. tundra* isolated from *Ae. caspius*) and PP475174 (*S. tundra* isolated from *Ae. vexans*). 

## 4. Discussion

This study represents a contribution to the distribution of *Dirofilaria* spp. and marks the first record of *S. tundra* in Serbia. It also provides valuable insight into the species of mosquito vectors and their host preference in this region. 

The previous paper published by Kurucz et al. [41] provided the first molecular evidence of *D. immitis* and *D. repens* nematodes from mosquito samples in Serbia. However, out of 73 locations, the present study only confirmed *D. immitis* in three locations in Vojvodina Province. Considering that the previous study treated a high number of *Cx. pipiens*, our study presented more information on other vectors such as *Ae. vexans*, *Ae. caspius*, *Ae. albopictus*, and *An. maculipennis* complex. European studies have confirmed infections by *D. immitis* in the following mosquito species: *Cx. pipiens* in Spain [49], Italy [33], and Turkey [50]; *Cx. theileri* in Madeira, Portugal [51], and on the Canary Islands, Spain [52]; *Ae. vexans* in Turkey [50,53]; and *Ae. albopictus*, *Ae. caspius*, *An. maculipennis*, and *Cq. richiardii* in Italy [33,54,55].

In our study, only *D. immitis* was detected in the analyzed mosquitoes collected at 73 locations. Although *D. repens* was earlier detected by Kurucz et al. [41], in this research, it was not found. All three positive samples in the present study belonged to *Cx. pipiens*. These mosquitoes were collected in three different locations (Svetozar Miletić, Glogonj, and Zrenjanin) that are not close to each other (Glogonj to Zrenjanin: 53 km, Svetozar Miletić to Zrenjanin: 135 km, and Glogonj to Svetozar Miletić: 172 km). Two of these locations are villages, and one is an urban settlement. The study of Kurucz et al. [41] detected these parasitic worms in mosquitoes at six locations, and their positive locations were also very distant. Bearing in mind that *Cx. pipiens* is a very bad flier [42], it is indicative that *D. immitis* is a widely spread parasite in Vojvodina Province. One location selected by Kurucz et al. [41] (Zrenjanin) overlaps with our results, demonstrating the persistent circulation of *D. immitis* in this city (from 2014 to 2021). 

Our analysis of blood-meal sources from mosquitoes collected in Zrenjanin and Glogonj (locations positive for *D. immitis*) demonstrated that three *Cx. pipiens* took blood from humans. Two mosquitoes were collected in Zrenjanin, and one was collected in Glogonj. In Zrenjanin, other blood meals were identified from the following animals: dog (in one mosquito), raven (in one mosquito), wild boar (in one mosquito), sparrow (in one mosquito), and pigeon (in two mosquitoes). In the location close to Zrenjanin, it was demonstrated that *Cx. pipiens* was feeding on a pigeon. These findings could also represent a contribution to our understanding of West Nile virus circulation, which is very frequently detected in these locations [56]. Interestingly, the blood meals of other collected mosquitoes (two *Cx. pipiens*) comprised cats’ blood. Earlier studies demonstrated that cats could become infected with *Dirofilaria* sp. but that it did not cause severe disease in them. Cats are not considered good hosts for *Dirofilaria* because the infections are attacked by their immune system before the nematodes can become adults [4]. It is estimated that the prevalence of feline infections in Europe is between 5 and 20% of the total canine prevalence in the same region [7].

The first systematic studies of dirofilariasis in dogs in Serbia were initiated at the beginning of the 21st century. One study was performed in Vojvodina Province and showed the endemic status of *D. repens* and *D. immitis* infection in dogs [35,36]. The climatic conditions in Serbia, coupled with the long activity periods of competent vectors such as *Cx. pipiens* [56] and *Ae. albopictus* [57], are considered suitable for the transmission of *D. immitis* and *D. repens* to humans and animals for at least half of the year (sometimes even more), depending on the air temperature [58,59]. The findings of Savić et al. [19] showed a prevalence of 26.30% for *D. immitis* infections in dogs, with 25.72% showing microfilariae. The prevalence of *D. repens* larvae was 1.45%. An earlier study showed a prevalence of 22.9% for *D. immitis*, while, for *D. repens*, it was 39.34% [36]. Several studies conducted in Serbia demonstrated an increasing trend of *D. immitis* infections and a decreasing trend of *D. repens* [19,35,60,61]. 

*Setaria tundra* is a new species on the list of parasites in Serbia. In this study, *S. tundra* was found in two locations (Iđoš and Mali Iđoš), which are almost 93 km apart. The records of *D. immitis* and *S. tundra* did not overlap in the same locations, possibly due to their different host preferences. *D. immitis*, which primarily infects domesticated dogs, is commonly found in areas closer to human settlements. In contrast, *S. tundra*, a parasite of deer, is expected to be present in locations near forests.

*Setaria nematodes* are classified into the Filarioidea superfamily, family Onchocercidae, and are parasites of different ungulates. At least four species of the genus Setaria are present in Europe: *S. equina* [62], *S. cervi* [63], *S. labiatopapillosa* [64], and *S. tundra* [65]. *Setaria tundra* was first described in Russia in 1928 [66], and, up to now, it has been reported in many European countries [67]. The reports from European countries are from (listed chronologically) Russia in 1928 [66], Austria in 1969 [68], Finland in 1970 [69], Sweden in 1973 [70], Norway in 1973 [71], Bulgaria in 1973 [72], Switzerland in 1974 [73], Germany in 1975 [74], Italy in 2003 [75], France in 2006 [76], Denmark in 2011 [77], Poland in 2010 [78], Hungary in 2013 [27], Spain in 2016 [79], Croatia in 2018 [24], and Slovakia in 2022 [25]. 

*Setaria tundra* isolated from *Ae. caspius* has shown similarity with *S. tundra* originally isolated from *Cq. richiardii* in Austria (MF695090), while *S. tundra* isolated from *Ae. vexans* has shown a similarity with *S. tundra* isolated from *Ae. vexans* in Hungary (KM452922). Based on the phylogenetic distance, we hypothesize that the two *S. tundra* isolates from Serbia originated from different geographic regions.

Olos et al. [23] hypothesized that the geographical expansion of *Setaria* nematodes may be indirectly related to wet and warm summers. This is because intermediate hosts are found in abundance, along with the high density of possible definitive hosts, as well as wild and domesticated ungulates. These authors stated that the recent focus on *S. tundra* has been due to its spreading range to the southern regions of Europe. This species of nematode has expanded its geographical range by hundreds of kilometers and is known to be a major cause of mass mortality in wild and semi-domesticated reindeer in Fennoscandia, Finland [80,81]. In northern Europe, the reindeer (*Rangifer tarandus*) is the major definitive host, yet the moose can serve as an asymptomatic carrier [65,82,83], while roe deer (*Capreolus capreolus*) and red deer (*Cervus elaphus*) serve as the definitive hosts in central and southern Europe [23,81,84]. In the review by Olos et al. [67], it is stated that domestic species such as sheep, goats, cattle, and horses are also potentially at risk [85,86,87,88]. Over the last decade, the populations of wild ruminants and wild boars have increased across Europe [89,90]. This expansion is accompanied by an apparent negative relationship between their abundance in the wild and their health status [91]. Considering that wild animals often enter cattle pastures and spread parasites to livestock, it is of great importance to maintain surveillance and control wildlife diseases [92]. 

This parasite can be transmitted by several species of mosquitoes, but particularly by those of the genus *Aedes* [2,93,94]. Microfilariae of this parasite have been reported in *Ae. vexans*, *Ae. caspius*, *Cx. pipiens*, *Culex torrentium* Matrini 1925, *Aedes annulipes* (Meigen, 1830), *Ae. sticticus*, *Aedes rossicus* Dolbeskin & Gorickaja 1930, and *Cq. richiardii,* in the following countries: Poland [95,96], Hungary [27,97], and Germany [22,26]. 

In the present study, *S. tundra* was found in Vojvodina Province in two analyzed mosquito species: *Ae. vexans* and *Ae. caspius*. When the blood meals of other mosquitoes from the same location and a nearby one were analyzed, the results showed that two mosquitoes of *Ae. vexans* were feeding on roe deer, and one had fed on a sheep. The DNA from the blood meal of *Ae. caspius* was not successfully identified. It is interesting to note that, upon analyzing the locations where these mosquitoes were collected, it was seen that their traps were not very near to the forests. One trap is located in the middle of a human settlement, while the other one is approximately one km away from the settlement. The second trap was actually placed between a field of sunflowers and a vineyard. Considering that the tested mosquitoes contained deer blood, we could offer two hypothesis. The first is that the mosquitoes may have flown from forested areas to locations near humans, as species like *Ae. vexans* and *Ae. caspius* are known to have good flight capacities and can travel long distances [42]. The second hypothesis is that the deer themselves moved closer to human settlements.

The number of analyzed mosquitoes did not yield a high number of positive cases of either *Dirofilaria* or *S. tundra*. Therefore, we cannot determine the prevalence. According to previous studies that focused on the detection of *Dirofilaria* in animals and humans, the expected positivity in mosquitoes was much higher than what was demonstrated in this study. It is necessary to perform a systematic screening of mosquitoes, at least in the locations with positive animals and humans, to better understand the prevalence and behavior/preferences of the parasite and to determine potential risks for human and animal populations. Future studies should be focused on analyzing the impact of environmental conditions on the spread and adaptation of these parasites in Serbia.

## 5. Conclusions

The present study provided an update on *D. immitis* in mosquitoes in Vojvodina Province and the Mačva region. Two new locations with *D. immitis* presence in vectors in Vojvodina were provided, along with the confirmation of the previously detected positive location where the circulation of the parasite is still active. *Setaria tundra* was detected on Serbian territory for the first time in this research. The analysis of blood meals provided insights into the preferences of the species that were positive for *Dirofilaria* and *S. tundra*. This opened many questions that can only be answered through systematic research into the identified hotspots, reservoirs, and detected mosquito vector species. 

## Figures and Tables

**Figure 1 animals-14-01255-f001:**
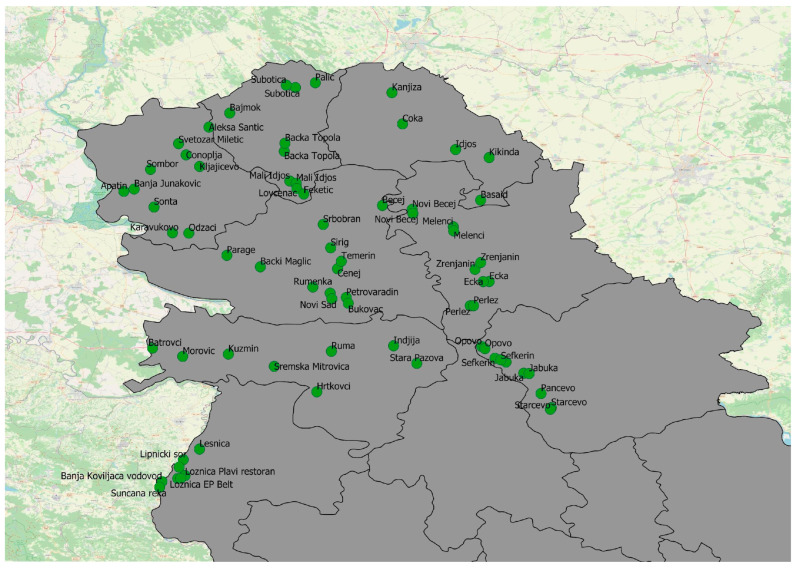
Sampling locations in Vojvodina Province and the Mačva region.

**Figure 2 animals-14-01255-f002:**
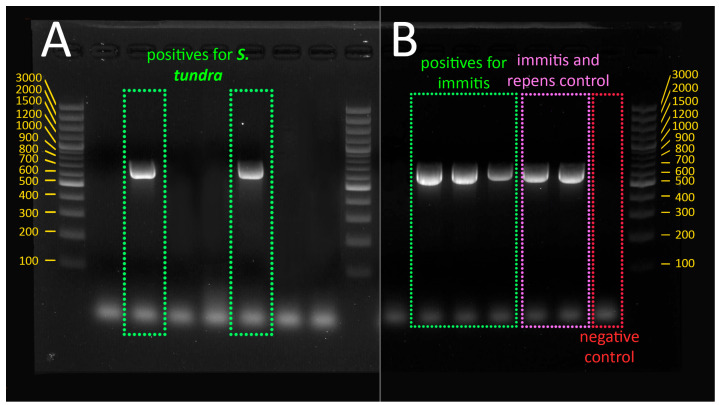
(**A**) *Ae. vexans* and *Ae. caspius* infected with *S. tundra* (product size of 650 bp, visualized on 1.3% agarose gel); (**B**) *Cx. pipiens* infected with *D. immitis* (product size of 650 bp, visualized on 1.3% agarose gel).

**Figure 3 animals-14-01255-f003:**
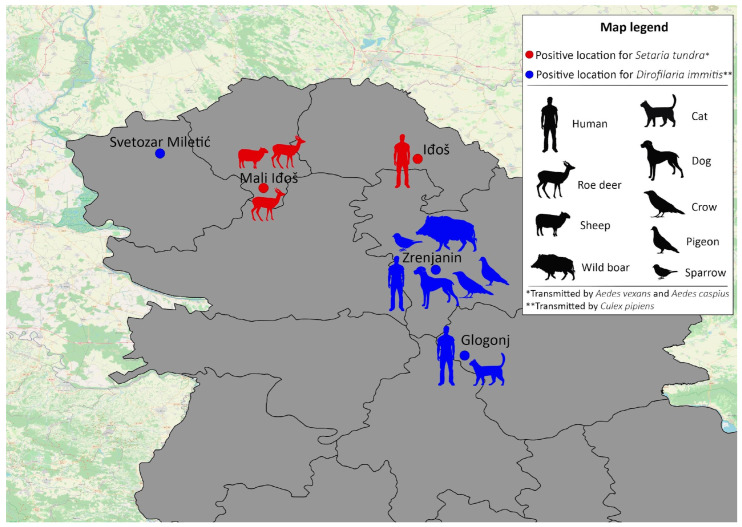
Locations with *Dirofilaria immits*- and *Setaria tundra*-positive mosquitoes, and detected blood-meal hosts.

**Figure 4 animals-14-01255-f004:**
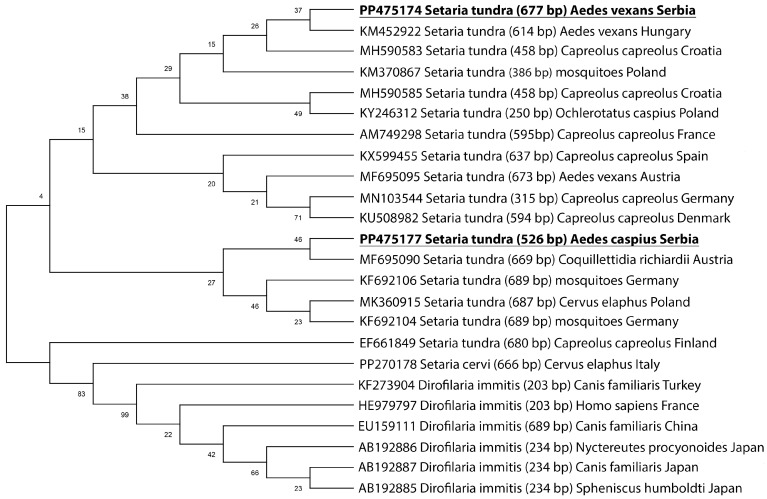
Maximum likelihood (ML) tree of the *Setaria* isolates identified in two mosquito species (*Ae. vexans* and *Ae. caspius*) from Serbia (bolded and underlined) and selected isolates from GenBank, based on a fragment of the cox1 gene. The numbers shown at the tree nodes represent bootstrap values based on 1000 replicates.

**Table 1 animals-14-01255-t001:** List of mosquito species, number of non-blood-fed and blood-fed female mosquitoes for each species, number of all analyzed specimens, and number of positive samples for *D. immitis* and *S. tundra*. All positive samples were in a pool of one mosquito per tube, except in the case of *Ae. vexans*, which had two mosquitoes in a single tube. En dash represents the lack of positive samples.

Mosquito Species	Number of		Total No. of Analyzed Mosquitoes	No. of Positive Samples for	
	Non-Blood-Fed	Blood-Fed		*D. immitis*	*S. tundra*
*Anopheles maculipennis*	383	15	398	–	–
*Aedes vexans*	1253	87	1340	–	1
*Aedes caspius*	305	11	316	–	1
*Aedes sticticus*	0	8	8	–	–
*Aedes albopictus*	580	0	580	–	–
*Culex pipiens*	0	225	225	3	–
*Culiseta annulata*	0	7	7	–	–
*Coquillettidia richiardii*	0	28	28	–	–
Total	2521	381	2902	3	2

## Data Availability

The data presented in this study are available in the article on the site of the MDPI journal *Animals*.

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
