# Peer review of "Dirofilaria sp. and Blood Meal Analysis in Mosquitoes Collected in Vojvodina and Mačva, and the First Report of Setaria tundra (Issaitshikoff & Rajewskaya, 1928) in Serbia"

_animals, 2024, doi:10.3390/ani14091255_

Round 1
Reviewer 1 Report
Comments and Suggestions for Authors
summary;
line 15 recently
Introduction
In the beginning (both abstract and introduction), there was no explanation regarding Setaria tundra. To enhance readership, please make an explanation about both dirofilaridae and setaria. Authors can add a brief introduction that Setaria is one of Onchocercidae family, which include both Setariinae and Dirofilariinae subfamily, etc.
Method
"Due to the low number of Aedes albopictus collected in 2021 and the high significance in filarial transmission of this invasive species"
"Traps were set up in the afternoon hours and operated overnight" Elaborate why the authors set up the traps during evening and then expect to capture aedes albopictus.
line 127-128 the taxonomy names were not uniformed; some were in brackets and some were not
line 178 Details of Bioedit was not mentioned (link or version)
line 197 Details of analysis such as BLAST (blastn/blastx/blastp? parameter used, searching set, program selection?) and phylogenetic construction (model, bootstrap no., length of nucleotide/protein sequence used?) were not mentioned in materials and methods
Results
Line 201-216 Will be helpful if these paragraphs were described using atable. Speficially regarding the detection of D. immitis, it was difficult to comprehend.
Line 201 by "analyzed" did the authors mean identified using pool? Since in the blood meal analysis, only 30 females were further individually analyzed for blood meal host (line 239)
line 238 In total,
line 248-260 paragraphs should not be italicized
line 250-254 detailed methods should be in method section or mentioned briefly in figure legend. Only describe the results in the result section.
Figure 4. There were only two Dirofilaria immitis sequences in the phylogenetic tree and mixed with other Setaria sequences rather than forming a separate clade. The same goes to Setaria cervi. This warrant more discussion of whether the previous articles mistakenly identified the sequence or the recently expanding setaria tundra discoveries.
Authors can add more dirofilaria sequences in the phylo tree and please also add the length of sequence used in the tree construction.
Discussion
Line 369 provide a reference to support statement
Author Response
Dear reviewer,
Thank you for reviewing our manuscript (animals-2950267) and providing suggestions for improvement. In response to your review, the following corrections have been implemented:
- Comment of Reviewer 1. Summary. line 15 recently
Authors’ response: The spelling has been corrected.
- Comment of Reviewer 1. Introduction. In the beginning (both abstract and introduction), there was no explanation regarding Setaria tundra. To enhance readership, please make an explanation about both dirofilaridae and setaria. Authors can add a brief introduction that Setaria is one of Onchocercidae family, which include both Setariinae and Dirofilariinae subfamily, etc.
Authors’ response: A brief introduction about Setaria tundra has been added as suggested both in abstract and in the introduction section.
- Comment of Reviewer 1. Method. "Due to the low number of Aedes albopictus collected in 2021 and the high significance in filarial transmission of this invasive species".
"Traps were set up in the afternoon hours and operated overnight" Elaborate why the authors set up the traps during evening and then expect to capture Aedes albopictus.
Authors’ response: The traps were set in the early afternoon hours, not in the evening. Therefore Ae. albopictus was still active at this time.
- Comment of Reviewer 1. line 127-128 the taxonomy names were not uniformed; some were in brackets and some were not
Authors’ response: Regarding the usage of brackets, we have used the standard system of scientific nomenclature writing. Meaning, if the original name of a species has changed at some point in scientific research, the original author who discovered the species will be written in parentheses. If the original name of a species has not been changed, the original author will not be in brackets.
- Comment of Reviewer 1. line 178 Details of Bioedit was not mentioned (link or version)
Authors’ response: The version of Bioedit has been added in parentheses.
- Comment of Reviewer 1. line 197 Details of analysis such as BLAST (blastn/blastx/blastp? parameter used, searching set, program selection?) and phylogenetic construction (model, bootstrap no., length of nucleotide/protein sequence used?) were not mentioned in materials and methods
Authors’ response: The suggestions for adding details for BLAST analysis is now applied both in “2.3. Identification of Dirofilaria sp.” and in “2.4. Identification of blood meal host”, as well as adding the phylogenetic construction in the materials and methods.
- Comment of Reviewer 1. Results. Line 201-216 Will be helpful if these paragraphs were described using a table. Speficially regarding the detection of D. immitis, it was difficult to comprehend.
Authors’ response: The table has been made and added as suggested.
- Comment of Reviewer 1. Line 201 by "analyzed" did the authors mean identified using pool? Since in the blood meal analysis, only 30 females were further individually analyzed for blood meal host (line 239)
Authors’ response: The numbers mean in total how many mosquitoes were analyzed for the parasite detection. The "analyzed" referred to both pools and individually tested.
- Comment of Reviewer 1. line 238 In total,
Authors’ response: The comma has been added.
- Comment of Reviewer 1. line 248-260 paragraphs should not be italicized
Authors’ response: Thank you. The correction has been made.
- Comment of Reviewer 1. line 250-254 detailed methods should be in method section or mentioned briefly in figure legend. Only describe the results in the result section.
Authors’ response: Lines 250-254 have been moved to the materials and methods section. Results remained as suggested.
- Comment of Reviewer 1. Figure 4. There were only two Dirofilaria immitis sequences in the phylogenetic tree and mixed with other Setaria sequences rather than forming a separate clade. The same goes to Setaria cervi. This warrant more discussion of whether the previous articles mistakenly identified the sequence or the recently expanding setaria tundra discoveries.
Authors’ response: Thank you for this interesting observation and the comment. After a thorough reconsideration of the phylogenetic analysis, the authors have revised the list of selected sequences. Two sequences were removed because they were not from the same gene as COI (cox1). As a result, we are submitting a newly analyzed phylogeny (phylogenetic tree), which is significantly more logical and accurate than the previous version.
- Comment of Reviewer 1. Authors can add more dirofilaria sequences in the phylo tree and please also add the length of sequence used in the tree construction.
Authors’ response: We have added more Dirofilaria sequences as suggested and added the length of sequence used in the tree construction.
- Comment of Reviewer 1. Discussion. Line 369 provide a reference to support statement
Authors’ response: The reference is added.
Reviewer 2 Report
Comments and Suggestions for Authors
The authors performed field collection of mosquitoes to check the presence of parasitic filarial nematodes in Vojvodina and Mačva, Serbia. Out of the 73 collection sites, they showed three positive for Dirofilaria immitis, and two positive for Setaria tundra which is the first time report in Serbia. Besides, these authors also tested the possible host preference of the mosquito by typing the blood fed mosquitoes. These results provided novel insights to a better understanding of the prevalence of parasitic nematodes as well as their potential mosquito vectors in Serbia. To this point, I would suggest this is an interesting and valuable filed study to be published in Animals. However, the data interpretation and discussion of the current form does not meet the scientific criteria, and should be carefully revised. The following major comments are for the further revision, regarding to data presentation, technical issue, as well as discussion.
Major comments:
1. It lacks a detailed explanation of the strategy of mosquito trap deployment. What is the criteria for the selection of deployment location?
2. One of the main aims of the current study is to explore the links between nematodes spread and environmental conditions as the authors mentioned. However, I found little information towards environmental conditions of the collection spots. It also lacks an good interpretation of the results (spread of positive mosquito) as well as possible mechanism discussion (animal and people density, or other environmental factors?).
3. The identification of the nematodes species is based on the 650bp GOI amplicon and subsequent sanger sequencing. It lacks the details of the GOI. Besides, is it a standard identification method? Considering it might be the first time to identify Setaria tundra in Serbia, it would be better to confirm the results, probably by another pair of primers?
4. According to the results, it seems that there is no overlap of the geographical distribution of the positive mosquitoes possess the two nematodes. The author should explain or discuss this in the Discussion part, at least to get rid of any potential detection or technical preference. Any other report showing the overlap of the two nematodes (in the same location of different mosquito pool?)
Other comments:
1. I suggest to move the Figure 1 to the Results part. Besides, it would be better to show the sample size in each location (e.g. the size of the dots).
2. Figure title is missing for Figure.2.
3. Line 250-260, check the fonts of the text.
4. A summary (e.g. a table) of the mosquito collections which are detected positive for filarial nematodes should be included in the main text. It should contain the pool strategy, mosquito species, sampling period and blood fed status.
Author Response
Dear reviewer,
Thank you for your comments. We have made the following changes to the manuscript (animals-2950267):
The authors performed field collection of mosquitoes to check the presence of parasitic filarial nematodes in Vojvodina and Mačva, Serbia. Out of the 73 collection sites, they showed three positive for Dirofilaria immitis, and two positive for Setaria tundra which is the first time report in Serbia. Besides, these authors also tested the possible host preference of the mosquito by typing the blood fed mosquitoes. These results provided novel insights to a better understanding of the prevalence of parasitic nematodes as well as their potential mosquito vectors in Serbia. To this point, I would suggest this is an interesting and valuable filed study to be published in Animals. However, the data interpretation and discussion of the current form does not meet the scientific criteria, and should be carefully revised. The following major comments are for the further revision, regarding to data presentation, technical issue, as well as discussion.
Major comments:
- Comment of Reviewer 2. It lacks a detailed explanation of the strategy of mosquito trap deployment. What is the criteria for the selection of deployment location?
Authors’ response: The authors added the sentence that explains the sampling criteria in the 2.1. Mosquito sampling and vector identification.
- Comment of Reviewer 2. One of the main aims of the current study is to explore the links between nematodes spread and environmental conditions as the authors mentioned. However, I found little information towards environmental conditions of the collection spots. It also lacks an good interpretation of the results (spread of positive mosquito) as well as possible mechanism discussion (animal and people density, or other environmental factors?).
Authors’ response: Thank you for your observation. The environmental factors are kept in the manuscript by mistake. The initial plan of this study was to include and analyses of environmental factors which influence spread and biology of the given parasite. Due to the incomplete data about the environmental factors, the authors decided not to include these data. Therefore, the answer to your comment is that the text is modified: environmental factors are excluded from the text, and we wrote a suggestion in the discussion section that future studies could perform this type of research.
- Comment of Reviewer 2. The identification of the nematodes species is based on the 650bp GOI amplicon and subsequent sanger sequencing. It lacks the details of the GOI. Besides, is it a standard identification method? Considering it might be the first time to identify Setaria tundra in Serbia, it would be better to confirm the results, probably by another pair of primers?
Authors’ response: Thank you for your observation. Sequencing of COI (cox1) amplicon following protocol described in Casiraghi et al. is a standard procedure allowing determination of species. In metabarcoding of animals, COI gene is frequently used as the marker of choice because no other genetic region can be found in taxonomically verified databases with sequences covering so many taxa. The results of the Sanger sequencing demonstrated the high identity of the samples in comparison with reliable data available in GenBank.
- Comment of Reviewer 2. According to the results, it seems that there is no overlap of the geographical distribution of the positive mosquitoes possess the two nematodes. The author should explain or discuss this in the Discussion part, at least to get rid of any potential detection or technical preference. Any other report showing the overlap of the two nematodes (in the same location of different mosquito pool?)
Authors’ response: Thank you for your comment. The authors added a sentence in the Discussion as suggested. We did not find overlapping in other studies. Studies are usually focused on either Dirofilaria or on Setaria.
Other comments:
- Comment of Reviewer 2. I suggest to move the Figure 1 to the Results part. Besides, it would be better to show the sample size in each location (e.g. the size of the dots).
Authors’ response: If possible, the authors prefer to leave the Fig 1 in Material and Method because it shows only the sampling spots and does not represent any result.
- Comment of Reviewer 2. Figure title is missing for Figure.2.
Authors’ response: Figure 2 has been added in the text accordingly.
- Comment of Reviewer 2. Line 250-260, check the fonts of the text.
Authors’ response: Correction has been made accordingly.
- Comment of Reviewer 2. A summary (e.g. a table) of the mosquito collections which are detected positive for filarial nematodes should be included in the main text. It should contain the pool strategy, mosquito species, sampling period and blood fed status.
Authors’ response: The table has been made and added as suggested. Pool strategy is given in the text, mosquito species, sampling period and blood fed status are given in the Supplementary table because it would not fit within the format of the journal page.
Reviewer 3 Report
Comments and Suggestions for Authors
Dear Authors,
the article is generally clear, well structured and well written and it is suitable for publication. I have only recognized a few minor issues to be addressed that I list below:
Lines 83-85: “Between 2006 and 2007 the reported prevalence for D. immitis was 7.2% in the Vojvodina and 3.2% in Branicevo regions [26, 27]. In the region of Belgrade, a few years later, the prevalence was 22.01%, with 3.97% of dogs showing co-infections with D. repens [10]
These percentages should be better explained. In the first sentence it should be specified that, probably, prevalence refers to tested dogs. In the second, the percentages seem to refer to the cases of co-infection of the two Dirofilaria species but, if this is true, why is information regarding the prevalence of single Dirofilaria species lacking?
Lines 250-260: check the use of italics
Lines250-254: please verify whether this information regarding methods can be moved to Methods section
Line 293: “Bearing in mind that Cx. pipiens is a very bad flier”
Please add a reference
Lines 297-288: “…Cx. pipiens took the blood (at least the last blood meal) from humans, two mosquitoes being from Zrenjanin and one from Glogonj”.
Please check language for clarity
Line 301: “including a dog, raven, wild boar, sparrow and pigeon (2 times)”
Please check language
Line 344: please add the scientific name also for the roe deer
Line 368-9: “Ae. vexans and Ae. caspius 368 have good flight capacities and can fly long distances”
Please add references
Line 382: “confirmed”
Since this is the first record, I should use a different term
Best regards
Comments on the Quality of English Language
Language is generally fine.
There are only a few sentences to be revised for clarity (listed in the comments to the Authors)
Author Response
Dear reviewer,
Thank you for reviewing our manuscript (animals-2950267) and providing suggestions for improvement. In response to your review, the following corrections have been implemented:
- Comment of Reviewer 3. Lines 83-85: “Between 2006 and 2007 the reported prevalence for D. immitis was 7.2% in the Vojvodina and 3.2% in Branicevo regions [26, 27]. In the region of Belgrade, a few years later, the prevalence was 22.01%, with 3.97% of dogs showing co-infections with D. repens [10]
These percentages should be better explained. In the first sentence it should be specified that, probably, prevalence refers to tested dogs. In the second, the percentages seem to refer to the cases of co-infection of the two Dirofilaria species but, if this is true, why is information regarding the prevalence of single Dirofilaria species lacking?
Authors’ response: Thank you for your observation. The sentence is improved. We hope that it is clearer now. The first study did not show any coinfection and that is why we did not mention in the first sentence.
- Comment of Reviewer 3. Lines 250-260:check the use of italics
Authors’ response: The mistake is corrected.
- Comment of Reviewer 3. Lines 250-254:please verify whether this information regarding methods can be moved to Methods section.
Authors’ response: The change is made accordingly.
- Comment of Reviewer 3. Line 293: “Bearing in mind that Cx. pipiens is a very bad flier” Please add a reference
Authors’ response: The reference is added.
- Comment of Reviewer 3. Lines 297-288: “…Cx. pipiens took the blood (at least the last blood meal) from humans, two mosquitoes being from Zrenjanin and one from Glogonj”. Please check language for clarity
Authors’ response: The sentence is improved.
- Comment of Reviewer 3. Line 301: “including a dog, raven, wild boar, sparrow and pigeon (2 times)” Please check language
Authors’ response: The sentence is simplified.
- Comment of Reviewer 3.Line 344: please add the scientific name also for the roe deer
Authors’ response: The Latin name is added.
- Comment of Reviewer 3.Line 368-9: “Ae. vexans and Ae. caspius 368 have good flight capacities and can fly long distances” Please add references
Authors’ response: The reference is added.
- Comment of Reviewer 3. Line 382: “confirmed” Since this is the first record, I should use a different term
Authors’ response: Thank you very much. The authors fully agree. The “confirmed” is replaced with detected.
Round 2
Reviewer 2 Report
Comments and Suggestions for Authors
The authors have made corrections accordingly, I have no other comments.